# Distinct catecholaminergic pathways projecting to hippocampal CA1 transmit contrasting signals during navigation in familiar and novel environments

**Chad Heer, Mark Sheffield\***

The Department of Neurobiology, The University of Chicago, Chicago, United States

## eLife assessment

This study presents **important** findings on the differential activity of noradrenergic and dopaminergic input to dorsal hippocampus CA1 in head-fixed mice traversing a runway in a virtual environment that is familiar or novel. The data are rigorously analysed, and the observed divergence in the dynamics of activity in the dopaminergic and noradrenergic axons is **solid**. Future studies, using specific manipulations of the two distinct midbrain inputs combined with behavioral testing, are required to strengthen the claim that distinct signals to the hippocampus cause distinct behavioral effects.

**\*For correspondence:**
sheffield@uchicago.edu

**Competing interest:** The authors declare that no competing interests exist.

**Abstract** Neuromodulatory inputs to the hippocampus play pivotal roles in modulating synaptic plasticity, shaping neuronal activity, and influencing learning and memory. Recently, it has been shown that the main sources of catecholamines to the hippocampus, ventral tegmental area (VTA) and locus coeruleus (LC), may have overlapping release of neurotransmitters and effects on the hippocampus. Therefore, to dissect the impacts of both VTA and LC circuits on hippocampal function, a thorough examination of how these pathways might differentially operate during behavior and learning is necessary. We therefore utilized two-photon microscopy to functionally image the activity of VTA and LC axons within the CA1 region of the dorsal hippocampus in head-fixed male mice navigating linear paths within virtual reality (VR) environments. We found that within familiar environments some VTA axons and the vast majority of LC axons showed a correlation with the animals' running speed. However, as mice approached previously learned rewarded locations, a large majority of VTA axons exhibited a gradual ramping-up of activity, peaking at the reward location. In contrast, LC axons displayed a pre-movement signal predictive of the animal's transition from immobility to movement. Interestingly, a marked divergence emerged following a switch from the familiar to novel VR environments. Many LC axons showed large increases in activity that remained elevated for over a minute, while the previously observed VTA axon ramping-to-reward dynamics disappeared during the same period. In conclusion, these findings highlight distinct roles of VTA and LC catecholaminergic inputs in the dorsal CA1 hippocampal region. These inputs encode unique information, with reward information in VTA inputs and novelty and kinematic information in LC inputs, likely contributing to differential modulation of hippocampal activity during behavior and learning.

## Introduction

Catecholamines have a well-established role in hippocampal function. Both dopamine and norepinephrine have been shown to impact hippocampal-dependent learning and memory (*de Silva et al., 2012*; *Retailleau and Morris, 2018*; *Tsetsenis et al., 2019*; *Hansen and Manahan-Vaughan, 2014*;

*Gibbs and Summers, 2002*; *Thomas, 2015*; *André et al., 2015*), alter synaptic plasticity (*Hagena and Manahan-Vaughan, 2013*; *Hansen and Manahan-Vaughan, 2014*; *Chu et al., 2011*; *Edison and Harley, 2012*; *Hagena and Manahan-Vaughan, 2012*; *Goh and Manahan-Vaughan, 2013*), modulate cell excitability (*Edelmann and Lessmann, 2018*; *Segal et al., 1991*), and influence the formation and stability of place cells (*Kentros et al., 2004*; *Retailleau and Morris, 2018*), hippocampal neurons that selectively fire at specific locations in an environment (*O'Keefe and Dostrovsky, 1971*). Traditionally, the main source of dopamine to the dorsal hippocampus was thought to be sparse inputs from the ventral tegmental area (VTA), while locus coerulues (LC) inputs provided the main source of norepinphrine.

VTA Dopaminergic (DA) inputs to dorsal CA1 of the hippocampus mainly innervate stratum oriens (*Takeuchi et al., 2016*; *Adeniyi et al., 2020*; *Adeyelu and Ogundele, 2023*), and their activity bidirectionally modulates Schaffer Collateral (CA3–CA1) synapses (*Rosen et al., 2015*), enhances persistence of reward-location associations (*McNamara et al., 2014*), and drives place preference (*Mamad et al., 2017*). VTA-hippocampus input activity can also bias place field location (*Mamad et al., 2017*), improve place field stability across days (*McNamara et al., 2014*), and drive reward expectation dependent enhancement of place field quality (*Krishnan et al., 2022*). However, many of the effects of DA modulation of the hippocampus have now been attributed to LC inputs as their activity enhances the strength of Schaffer Collateral synapses (*Takeuchi et al., 2016*), improves memory retention (*Kempadoo et al., 2016*), improves place field stability across days (*Wagatsuma et al., 2018*), and can bias place fields to a location when paired with a reward (*Kaufman et al., 2020*) through DA mechanisms. Although many of the effects of LC and VTA are overlapping, potentially indicating shared mechanisms of action, they are believed to play different roles in spatial learning and memory (*Duszkiewicz et al., 2019*). LC inputs influence the encoding of novel environments (*Kempadoo et al., 2016*; *Wagatsuma et al., 2018*), while VTA DA inputs increase persistence of reward context associations (*McNamara et al., 2014*), alter hippocampal neurons firing rate (*Adeniyi et al., 2020*), and their suppression can evoke place avoidance (*Mamad et al., 2017*). It is possible that these differences arise because of the differences in activity observed between LC and VTA DA neurons. Therefore, characterizing the encoding properties of LC and VTA inputs directly in the hippocampus during navigation and spatial learning would provide important insights into the specific roles of these distinct neuromodulatory pathways.

Additionally, recent findings indicate considerable heterogeneity in the activity of VTA (*Engelhard et al., 2019*) and LC (*Uematsu et al., 2017*; *Noei et al., 2022*; *Chandler et al., 2014*) neurons, highlighting the need for projection specific recordings. Therefore, we functionally imaged VTA DA and LC axons with single-axon resolution in dCA1 of mice as they navigated familiar and novel virtual reality (VR) environments for rewards. We observed distinct encoding properties between these sets of inputs during navigation and in response to environmental novelty. During the approach to a previously rewarded location, the activity of most VTA DA axons ramped up. In contrast, most LC axons did not show this ramping activity and instead predicted the start of motion. Additionally, a majority of LC axons and some VTA axons showed activity associated with the animal's velocity. Following exposure to a novel environment, VTA axon ramping-to-reward signals greatly reduced but LC axon activity sharply increased. These findings support distinct roles for VTA and LC inputs to the hippocampus in spatial navigation of rewarded and novel environments.

## Results

To record the activity of dopaminergic inputs to the dorsal hippocampus, we expressed axon-GCaMP6s or axon-GCaMP7b in LC or VTA neurons of different mice. We utilized the NET-Cre mouse line (*Wagatsuma et al., 2018*) to restrict expression to noradrenergic LC neurons, and the DAT-Cre line (*Zhuang et al., 2005*) to restrict expression to dopaminergic VTA neurons (*Figure 1b*). Mice were then head-fixed and trained to run a linear VR track for water rewards delivered through a stationary waterspout when they reached the end of the virtual track (*Figure 1a*). Following reward delivery, mice were teleported to the beginning of the track and allowed to complete another lap. On experiment day mice navigated the familiar, rewarded VR environment for 10 min while two-photon microscopy was used to image the calcium activity of LC (87 regions of interest [ROIs] from 22 imaging sites in 16 mice) or VTA (9 ROIs from 8 imaging sites in 8 mice) axons in the dorsal CA1 (*Figure 1c*). Based on the z-axis depth of the recording planes, and the presence of increased autofluorescence in

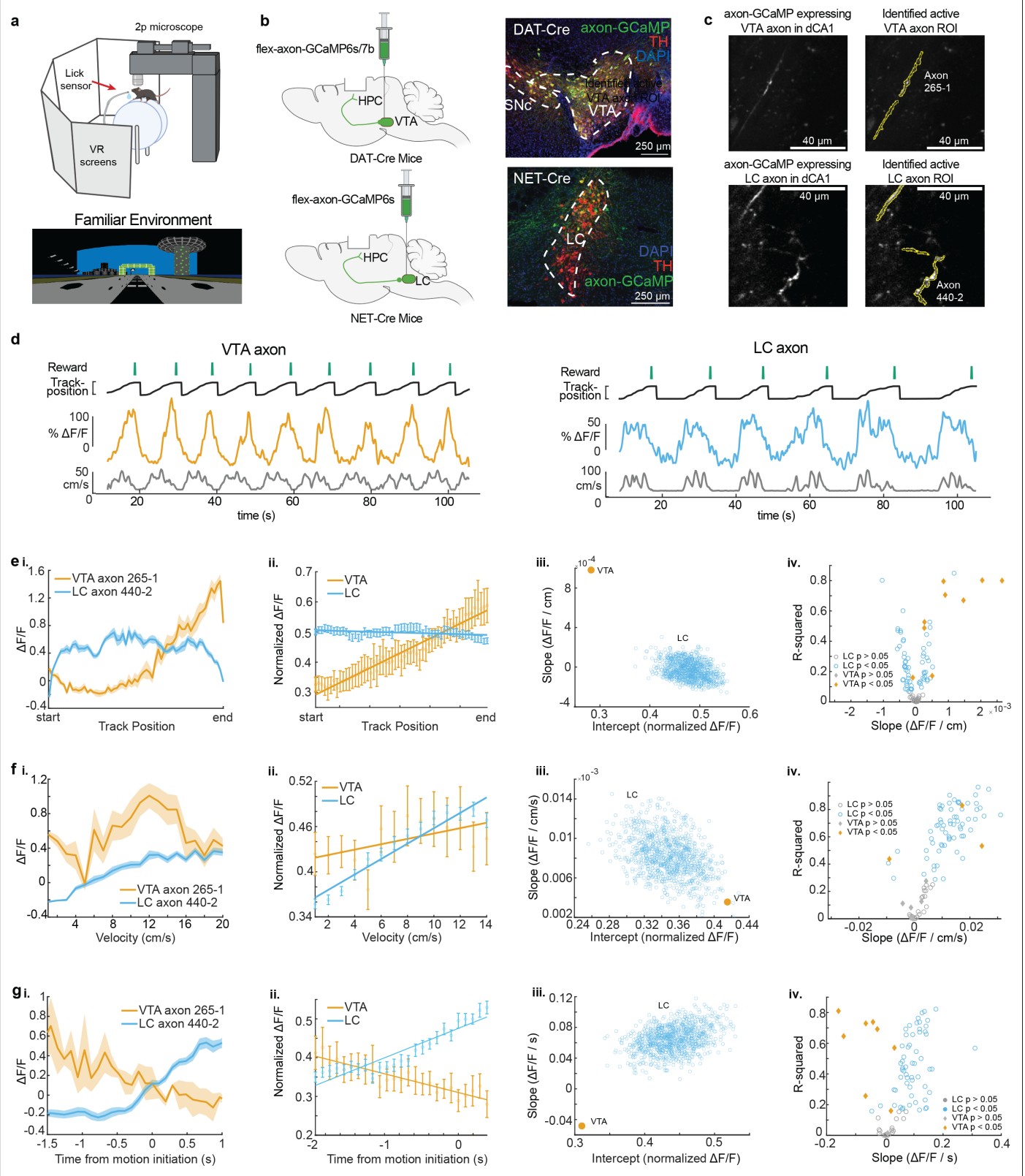

**Figure 1.** Distinct activity dynamics in ventral tegmental area (VTA) and locus coeruleus (LC) axons during navigation of familiar environments. (**a**) Experimental setup (top), created with BioRender.com. Example virtual reality environment. (**b**) Schematic representation of injection procedure (left). Representative coronal brain sections immunostained for tyrosine hydroxolase (TH) from a DAT-Cre mouse showing overlapping expression of axon-GCaMP (green) and TH (red) in VTA neurons (top) and from a NET-Cre mouse showing overlapping expression of axon-GCaMP (green) and TH (red)

*Figure 1 continued on next page*

*Figure 1 continued*

in LC neurons (bottom). (**c**) Example CA1 field of view of VTA axons (top) and LC axons (bottom). Extracted regions of interest (ROIs) used for example VTA and LC activity throughout the figure. (**d**) Example DAT-Cre mouse (left) and NET-Cre mouse (right) with aligned reward delivery (top, green), mouse track position (black), $\Delta F/F$ from example ROI (VTA – orange, LC – blue), and mouse velocity (bottom, gray). (**e, i**) Position binned $\Delta F/F\pm$ s.e.m in example VTA (orange) and LC (blue) ROIs during navigation of the familiar rewarded environment. (**ii**) Population average position binned $\Delta F/F\pm$ s.e.m. in VTA ROIs (orange, $n$ = 9 ROIs from 8 mice) and LC ROIs (blue, $n$ = 87 ROIs from 27 sessions in 17 mice) . Linear regression analysis (on all data points, not means) shows that the population of VTA ROIs increase activity during approach of the end of the track while the population of LC ROIs have consistent activity throughout all positions. Linear regression, $F$ test, VTA, $p < 1e − 21$, LC, $p < 0.01$. (**iii**) The LC dataset was resampled 1000× using $n$ = 9 ROIs to match the number of VTA ROIs and the slope and intercept of the regression line were measured each time (blue dots). The VTA slope is steeper than all LC slopes indicating a stronger positive relationship between position and activity for VTA axons. (**iv**) Linear regression of position binned activity of individual VTA (orange diamonds) and LC (blue, circles) ROIs. The majority (8/9) of VTA ROIs show a significant positive relationship with position while LC ROIs show both a positive (25/87) and negative (37/87) relationship. (**F, i**) Same example ROIs as (**d**) binned by velocity. (**ii**) Same data as (d, ii,) binned by velocity. Linear regression shows that the population of VTA and LC ROIs have a significant relationship with velocity. Linear regression, $F$ test, VTA, $p < 0.05$, LC, $p < 1e − 68$. (**iii**) Resampling shows the VTA slope and intercept is within the resampled LC slopes and intercepts indicating similar relationships with velocity. (**iv**) Linear regression of individual VTA and LC axons shows the majority (63/87) of LC ROIs have a significant positive relationship with velocity while only two VTA ROIs show this relationship. (**g, i**) Same example ROIs as (**d**) aligned to motion onset. (**ii**) Same data as (d, ii) aligned to motion onset. Linear regression shows that the population of VTA axons have a negative slope prior to motion onset while LC axons have positive slope. Linear regression, $F$ test, VTA, $p < 0.01$, LC, $p < 1e − 65$. (**iii**) Resampling shows the VTA slope is negative while all resampled LC slopes are positive. (**iv**) Linear regression of individual VTA and LC ROIs shows the majority (56/87) of LC ROIs have a significant positive slope prior to motion onset while the majority (6/9) of VTA ROIs have a negative slope.

The online version of this article includes the following source data and figure supplement(s) for figure 1:

**Source data 1.** Fluorescence data of ventral tegmental area (VTA) and locus coeruleus (LC) axons in familiar virtual reality (VR) environments.

**Figure supplement 1.** LC and VTA axon activity as a function of time and distance to reward.

**Figure supplement 1—source data 1.** Fluorescence data for ventral tegmental area (VTA) and locus coeruleus (LC) axons aligned by time and distance to reward.

**Figure supplement 2.** VTA DA axons expressing axon-GCaMP6s or axon-GCaMP7b show the same trends as a function of behavioral variables.

**Figure supplement 2—source data 1.** Fluorescence data of GCaMP6s and GCaMP7b ventral tegmental area (VTA) axons.

**Figure supplement 3.** Distinct activity dynamics in VTA DA axons expressing axon-GCaMP6s and LC axons expressing axon-GCaMP6 during navigation of familiar environments.

**Figure supplement 3—source data 1.** Fluorescence data of GCaMP6s ventral tegmental area (VTA) and locus coeruleus (LC) axons in familiar virtual reality (VR) environments.

stratum pyramidal, we determined all 9 VTA axons were in *Stratum Oriens*, while for LC recordings, 18 sessions (78 axons in 11 mice) occurred in *Stratum Oriens* and 5 sessions (9 axons in 5 mice) in *Stratum Pyramidalis*. Example VTA (left, orange) and LC (right, blue) axon calcium activity aligned to the animal's behavior are shown in *Figure 1d*. Axons from both brain regions showed periodic activity linked to the animals' exploration of the VR environment.

## Distinct activity dynamics in VTA and LC inputs during rewarded navigation of a familiar environment

To examine axon activity further, we first looked at the mean activity across all axons as a function of normalized track position (*Figure 1e*). As previously reported (*Krishnan et al., 2022*), VTA DA axons increase activity along the track, peaking at the reward location at the end of the track. In contrast, LC input activity remains relatively constant across all positions along the track (*Figure 1e,iii*). To examine if this difference could be due to the lower sample size of VTA axons compared to LC axons, the LC axons were down-sampled to match the VTA sample size ($n$ = 9 regions of interest [ROIs] in 8 mice) and the slope and intercepts of the down-sampled data were found. This was repeated 1000 times and did not generate any LC data points that overlap with VTA data demonstrating the difference in relationship between position and activity was not due to the different sample sizes (*Figure 1e,iii*). We also examined the position related activity of individual VTA and LC axons and observed a positive relationship between position and activity in 88.9% of VTA ROIs (8/9 in 7 mice) but only 28.7% of LC ROIs (25/87 from 12 sessions in 12 mice) while 42.5% of LC ROIs (37/87 from 16 sessions in 9 mice) had a negative relationship between position and activity (*Figure 1e,iv*). To account for the different track lengths between VTA and LC recordings, we also looked at the virtual distance from the rewarded end of the track as well as time from reward (*Figure 1—figure supplement 1*). The same trends were

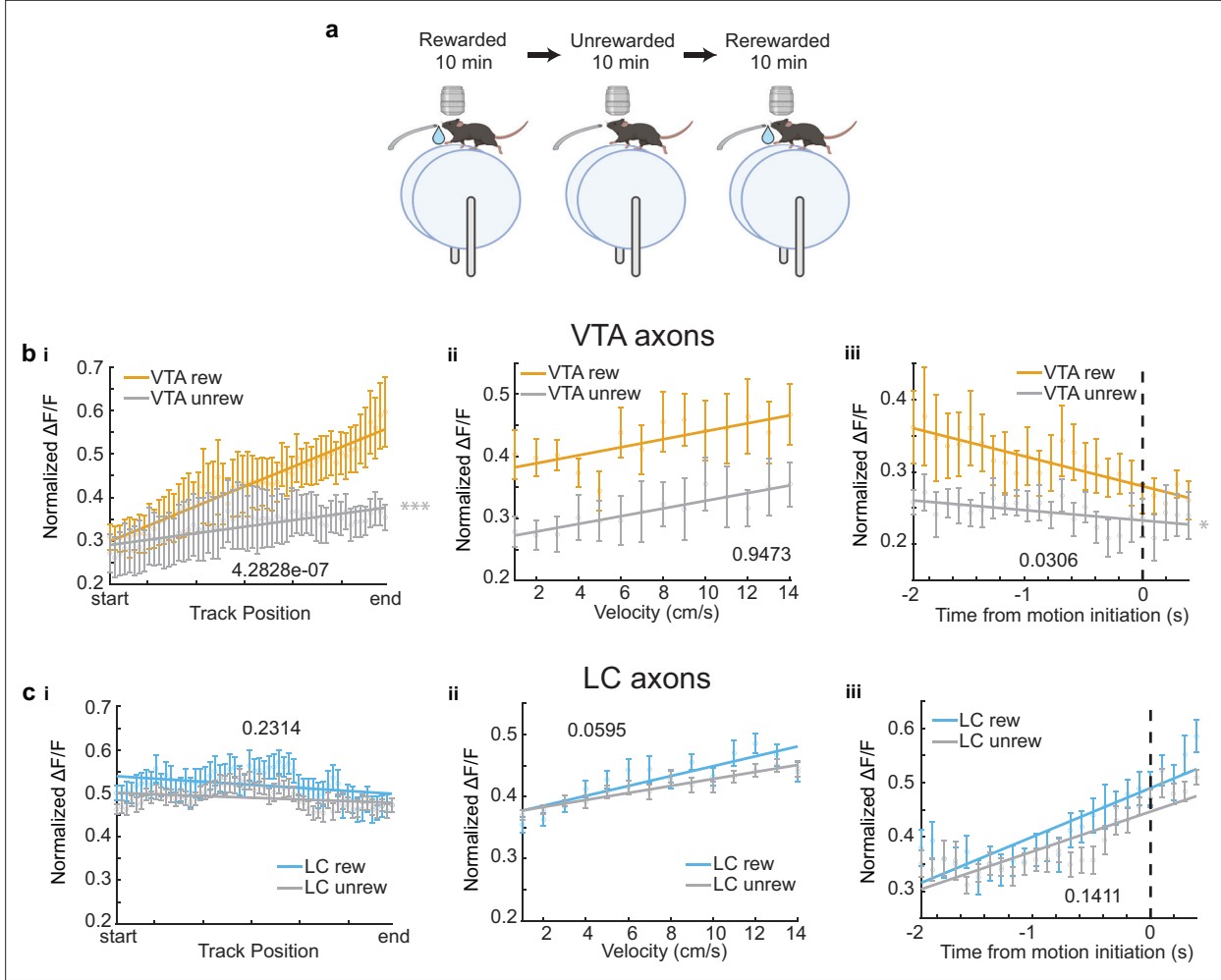

**Figure 2.** Removal of reward restructures ventral tegmental area (VTA) but not locus coeruleus (LC) input activity during spatial navigation. (**a**) Experimental paradigm. (**b, i**) Population average position binned $\Delta F/F \pm$ s.e.m. of VTA regions of interest (ROIs) ($n$ = 6 ROIs in 6 mice) in the rewarded (VTA rew, orange) and unrewarded conditions (VTA unrew, gray). Linear regression, $F$ test, Rewarded, $p < 1e - 22$, Unrewarded, $p < 0.001$. (**ii**) Same data as (**b, i**) binned by velocity. Linear regression, $F$ test, Rewarded, $p < 0.05$, Unrewarded, $p < 0.05$. (**iii**) Same data as (**b, i**) aligned to motion onset. Linear regression, $F$ test, Rewarded, $p < 1e - 4$, Unrewarded, $p < 0.05$. (**c, i**) Population average position binned $\Delta F/F \pm$ s.e.m. of LC ROIs ($n$ = 26 ROIs in 7 sessions in 4 mice) in the rewarded (LC rew, blue) and unrewarded conditions (LC unrew, gray). Linear regression, $F$ test, Rewarded, $p < 0.01$, Unrewarded, $p < 0.01$. (**ii**), Same data as (**c, i**) binned by velocity. Linear regression, $F$ test, Rewarded, $P < 1e - 12$, Unrewarded, $p < 1e - 15$. (**iii**) Same data as (**c, i**) aligned to motion onset. Linear regression, $F$ test, Rewarded, $p < 1e - 21$, Unrewarded, $p < 1e - 26$. The slope of each unrewarded measure was compared to the familiar laps using a one-way ANCOVA with Tukey HSD post hoc test. *$p < 0.05$, ***$p < 1e - 4$. This figure was created with BioRender.com.

The online version of this article includes the following source data for figure 2:

**Source data 1.** Fluorescence data of ventral tegmental area (VTA) and locus coeruleus (LC) axons in rewarded and unrewarded virtual reality (VR) environments.

seen for VTA and LC axons, with 8 VTA ROIs showing positive relationships with both distance and time from reward, and 21 LC ROIs showing a positive relationship with distance from reward and 31 LC ROIs showing a positive relationship with time to reward. This suggests that track length does not influence the encoding properties of VTA and LC axons.

For a subset of VTA ($n$ = 6) and LC ROIs ($n$ = 26), the reward at the end of the track was removed, and the activity of these axons in the unrewarded condition was examined (*Figure 2a*). While the slope of the VTA population activity across positions significantly decreased (*Figure 2b,i*) as expected and previously reported (*Krishnan et al., 2022*), the LC population activity across positions did not significantly change (*Figure 2c,i*). This confirmed that the ramping activity in VTA axons was due to

the animal's proximity to an expected reward and this signal was not present in the average activity of LC axons .

Next, we investigated the mean activity of these axons as a function of velocity. The population mean of both VTA and LC axons increased as velocity increased (*Figure 1f*). This is consistent with the finding that LC inputs to dCA1 encode velocity (*Kaufman et al., 2020*) and the finding that some DA VTA neurons encode kinematics (*Engelhard et al., 2019*). Again, to account for differences in sample size, we down-sampled the LC axons 1000 times and found the slope and *y*-intercept of each sampling. The overlap of the VTA and LC slopes and intercepts confirms we cannot conclude any differences in velocity-related activity in the VTA and LC axon populations (*Figure 1f,iii*). However, analyzing individual ROIs, we observed a statistically significant positive relationship between velocity and activity in the majority, 72.4%, of LC axons (63/87 ROIs from 21 sessions in 13 mice), while only 28.6% (2/9 ROIs in 2 mice) of VTA axons showed a positive relationship with velocity (*Figure 1f,iv*). The strong velocity correlated activity in a small subset of VTA DA axons indicates heterogeneity in the activity of these inputs similar to what is observed in VTA DA somas (*Engelhard et al., 2019*). To determine if the velocity correlated activity in VTA DA axons is confounded by the ramping-to-reward activity, we examined this activity in the unrewarded condition where the ramping-to-reward activity is absent. Here, the VTA population activity across all velocities is decreased but the slope, or the relationship between velocity and activity, remains unchanged (*Figure 2b,ii*). This is consistent with a subset of VTA axons encoding velocity information, while the overall decrease in activity is explained by the decrease in reward related activity demonstrated in *Figure 2b,i*. However, the relationship between velocity and LC activity is unchanged in the unrewarded environment (*Figure 2c,ii*), indicating condition invariant velocity encoding in LC axons.

Finally, we examined the activity of LC and VTA axons during rest and the transition to movement. The population of LC axons ramped up in activity during the 2 s leading up to motion onset (*Figure 1g*). This is consistent with reports of activity of LC axons in cortical areas (*Reimer et al., 2016*) showing LC activity prior to motion onset. In contrast, VTA axons show decreasing activity during the 2 s leading up to motion onset (*Figure 1g*). This ramping down in VTA axon activity is likely due to most periods of immobility occurring between reward delivery and the start of the next lap, during which we previously demonstrated reward related activity decays in VTA axons (*Krishnan et al., 2022*). Indeed, in the unrewarded condition the negative slope of the VTA population activity leading up to motion onset disappears (*Figure 2b,iii*), indicating this relationship is largely driven by the presence of reward rather than motion onset. However, the LC population activity remains unchanged in the unrewarded environment (*Figure 2c,iii*), suggesting this activity is related to the lead up to motion initiation. These differences in activity are not an artifact of lower sample size of VTA ROIs as shown by down-sampling the LC ROIs activity 1000 times and measuring the slopes and intercepts of the down-sampled data did not generate any data points that overlapped with the VTA slope and intercept (*Figure 1g,iii*). In further support of distinct activity profiles leading up to motion onset, we found that the majority (6/9 ROIs in 6 mice) of VTA ROIs decreased in activity leading up to motion onset but only 2/87 LC ROIs in one session decreased in activity, while the majority (56/87 ROIs in 17 sessions in 9 mice), of LC ROIs increased in activity leading up to motion onset (*Figure 1g,iv*). Together, this analysis demonstrates overlapping but distinct activity in VTA and LC axons during spatial navigation with VTA axons showing strong activity correlated with distance to reward and some velocity correlated activity, while LC axons demonstrate activity correlated to velocity and time to motion onset.

Because the DAT-Cre mice were injected with a virus for expression of either axon-GCaMP6s (four mice) or axon-GCaMP7b (four mice), we asked whether these two GCaMP variants led to different activity dynamics (*Figure 1—figure supplement 2*). The axon-GCaMP6s (five ROIs) and axon-GCaMP7b (four ROIs) both had a positive relationship with position, negative relationship with time to motion initiation, and a neutral relationship with velocity, although the GCaMP6s relationships were stronger (*Figure 1—figure supplement 2a, c*). Importantly, when we compared only axon-GCaMP6s expressing VTA ROIs with the axon-GCaMP6s expressing LC ROIs (*Figure 1—figure supplement 3*), we saw similar relationships as the comparisons using all VTA ROIs (*Figure 1*).

## Environmental novelty induces activity in LC but not VTA inputs

Both LC and VTA neurons have been shown to respond to novel environments (*Takeuchi et al., 2016*). Therefore, we aimed to examine the activity of VTA DA and LC inputs to the hippocampus during

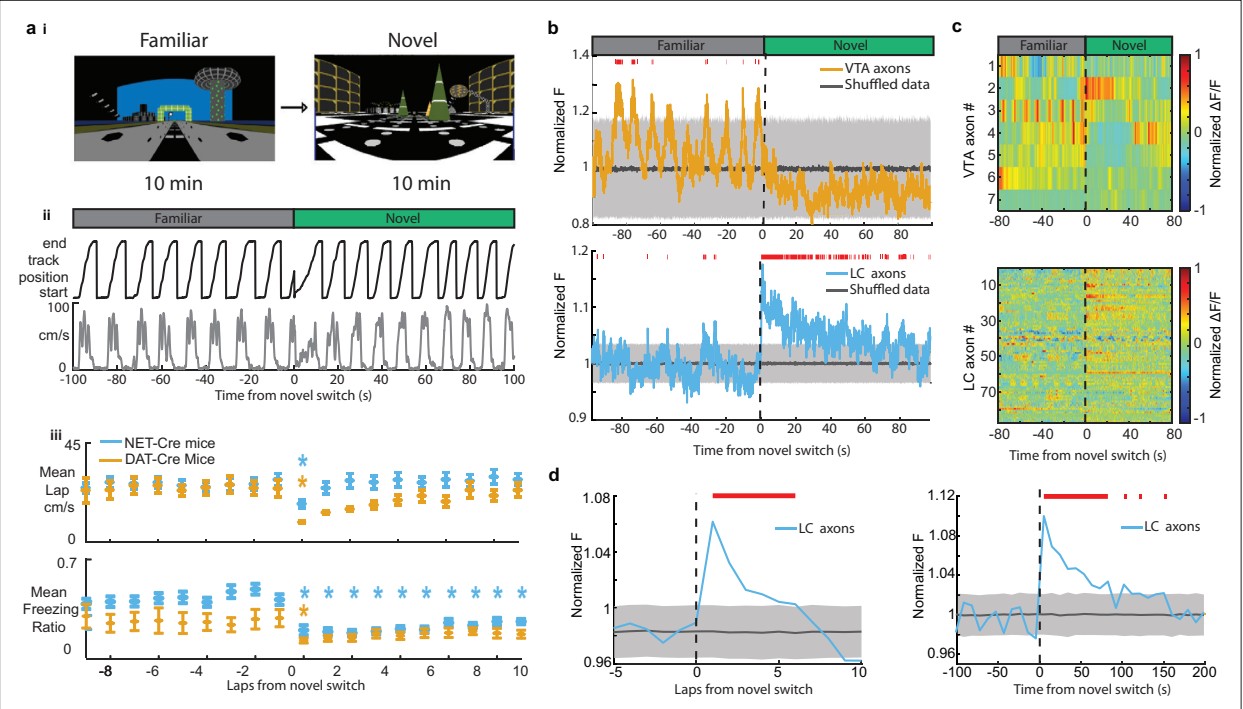

**Figure 3.** Exposure to a novel environment causes an abrupt and sustained increase in activity in locus coeruleus (LC) but not ventral tegmental area (VTA) inputs to dCA1. (**a, i**) Experimental paradigm. (**ii**) Behavior from example mouse during the transition from the familiar virtual reality (VR) environment to a novel VR environment showing the animals track position (top, black) and velocity (bottom, gray). (**iii**) The average running velocity ± s.e.m. of all mice (n = 7 DAT-Cre mice; n = 17 NET-Cre mice) during the transition to a novel VR environment (top). The average freezing ratio of all mice ± s.e.m. calculated as the time spent immobile (velocity <5 cm/s) divided by the total lap time. Each lap was compared to the final lap in the familiar environment using a one-way ANOVA with Tukey HSD post hoc test (Blue *, NET-Cre mice $P < 0.05$; Orange *, DAT-Cre mice $P < 0.05$). (**b**) Mean normalized fluorescence of all VTA regions of interest (ROIs) (top, n = 7 sessions in 7 mice) and LC ROIs (bottom, n = 27 sessions in 17 mice) aligned to the switch to the novel environment. To define a baseline and 95% CI (gray shaded region), 1000 shuffles were created from the calcium traces and down-sampled to match the sample size and averaged. This was repeated 1000 times and the mean and 95% CI of this shuffled data were determined for each frame. Red lines indicate periods where two or more consecutive frames passed above the %CI of the shuffled baseline. (**c**) Normalized $\Delta F/F$ activity of all VTA ROIs (top) and LC ROIs (bottom) aligned to the switch to the novel VR environment. (**d**) The normalized fluorescence of all LC ROIs binned by lap (left) or into 50 frame bins (right). The baseline and 95% CI (gray shaded region) were defined using the same method as in (**b**) Red lines indicate bins above the baseline 95% CI.

The online version of this article includes the following source data and figure supplement(s) for figure 3:

**Source data 1.** Fluorescence data of ventral tegmental area (VTA) and locus coeruleus (LC) axons in novel virtual reality (VR) environments.

**Figure supplement 1.** Reward related activity in VTA DA axons is diminished in a novel environment.

**Figure supplement 2.** Increased LC axon activity following exposure to a familiar environment and during immobile periods in a novel environment.

exposure to novel VR environments. Following 10 min in the familiar environment, mice were teleported to a novel VR environment of the same track length, with a reward at the same position at the end of the track. Following teleportation, we found the running speed of both DAT-Cre and NET-Cre mice transiently decreased (*Figure 3a*) and they spend less time immobile (*Figure 3a*), demonstrating mice recognize they are navigating a novel environment. While the velocity quickly recovered for both groups of mice, the freezing ratio, or amount of time spent immobile, never recovered in the first 10 laps in the novel environment for the NET-Cre mice indicating some novelty-induced changes in behavior persist.

We aligned VTA and LC axon activity to the switch to the novel environment and investigated changes in activity due to exposure to novelty. To test whether the mean axon activity is significantly elevated or lowered, we defined a baseline by generating 1000 shuffles of the axon traces across the entire recording sessions, down-sampling the shuffled data 1000 times to match the VTA (n = 7) and LC (n = 87) sample sizes, and calculating the mean and 95% CI of the shuffled data. After teleportation, the periodic activity observed in the mean of VTA axons, likely reflecting ramping-to-reward

signals in each individual axon, disappeared (*Figure 3b*). This is evident in the traces of most of the individual VTA ROIs showing a loss of the ramping-to-reward signal (*Figure 3c)*, and in the VTA population position binned activity showing a significant reduction in ramping activity (Supp. Fig. 1). However, one VTA ROI showed an increase in activity immediately following exposure to novelty (*Figure 3c*), indicating heterogeneity across VTA axons in CA1 and the lack of a novelty signal on average may be due to a small sample size.

Strikingly, LC axons show a dramatic increase in mean activity that remained elevated for >1 min following exposure to the novel environment (*Figure 3b*) similar with findings that LC cell body activity is elevated for minutes following exposure to environmental novelty (*Takeuchi et al., 2016*). A significant increase in activity above baseline activity following the switch to a novel environment can be seen in 36 LC axon ROIs (from 15/22 sessions in 10/16 mice) (*Figure 3c*). To further characterize this activity, we found the mean population activity for each lap and separately for 10 s time bins leading up to and following exposure to the novel environment. This analysis shows that LC activity is significantly elevated above baseline for six consecutive laps and approximately 90 s following exposure to the novel environment (*Figure 3d*). These findings demonstrate that LC inputs signal environmental novelty, supporting a role for these inputs in novelty encoding in the hippocampus.

## Novelty-induced changes in behavior explain the late but not early increases in LC activity

It is possible that the change in the amount of time the animals spend running versus immobile in the novel environment could explain the increase in LC activity in the novel environment, as LC activity is related to behavior (*Figure 1*). For instance, LC axons show elevated activity during motion versus rest (*Figure 1g*). Therefore, an increase in the time spent in motion upon exposure to the novel environment could lead to an increase in LC activity. To account for the differences in time spent running vs immobile between the two environments, we removed any periods where the mice were immobile to isolate the effects of novelty from changes in behavior (*Figure 4b*). When only looking at activity during running in both environments, we found that LC axon activity is elevated for two laps, or 40 s, in the novel environment (*Figure 4b–e*). Additionally, when we look only at activity when the mice were immobile, we see that activity is elevated for 30 s in the novel environment (*Figure 3—figure supplement 2b*). Together, this indicates that there are two separate components that drive LC axon activity during the initial exposure to the novel environment. One, a shorter purely novelty-induced increase in activity which occurs during the first two laps, or about 40 s, in the novel environment. Two, a behavior-induced increase in LC activity caused by an increase in the percentage of time spent running that extends beyond the increase in the novelty-induced activity for six laps or 90 s.

If the short, novelty-induced signal in LC axons is an additional signal riding on top of the behavior correlated signals – position, velocity, and motion onset – we would expect a disruption of these behavioral correlations during the initial lap in a novel environment. To test this we examined these behavioral correlations lap-by-lap following exposure to the novel environment. Indeed, the slopes of the position binned, velocity binned, and motion onset aligned data are all significantly more negative in the first lap in the novel environment than the final laps of the familiar environment (*Figure 4f–h*). This is consistent with a decaying novelty signal that peaks at the start of the first lap and rides on top of these behaviorally correlated signals. This produces an elevation in activity at positions near the start of the first lap that is lower at positions near the end of the first lap, causing a negative slope relationship between position and LC axon activity on the first lap (*Figure 4f*; light green line). Furthermore, low velocities occur at the start of each lap compared to the end of the lap. Because the novelty signal is highest when animals are running slowest, the novelty signal flattens the velocity–LC activity relationship (*Figure 4g*; light green line). Lastly, rest periods typically occur at the start of the track. Therefore, motion onset encoding on the first lap in the novel environment occurs when the novelty signal is highest, again, flattening the relationship (*Figure 4h*; light green line). By the third lap in the novel environment, where the novelty-induced signal is no longer observable, the relationships between LC activity and position is no longer different than the relationship in the familiar environment (*Figure 4f*, green). The relationship between LC activity and motion onset is also no longer different by the third lap in the novel environment (*Figure 4h*). Although the relationship between velocity and LC activity is different in the third novel lap than that of the final familiar laps, by the final lap in the novel environment it is no longer different than the familiar relationship (*Figure 4g*). Together this demonstrates

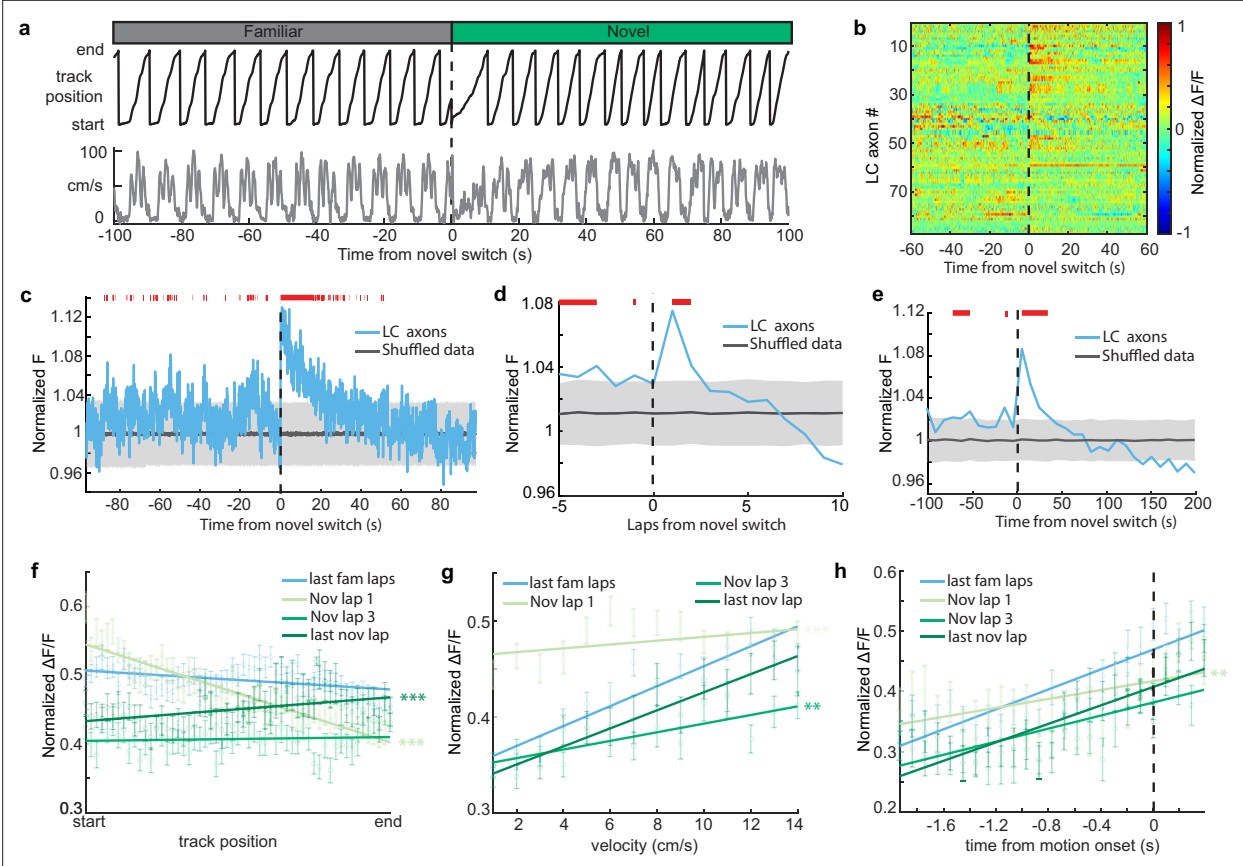

**Figure 4.** Novelty-induced changes in behavior explain the late but not early increases in locus coeruleus (LC) activity. (**a**) Good behavior from example mouse following removal of freezing periods (velocity <0.2 cm/s) during the transition from the familiar to a novel virtual reality (VR) environment showing the animals track position (top, black) and velocity (bottom, gray). (**b**) Normalized $\Delta F/F$ activity of all LC regions of interest (ROIs) aligned to the switch to the novel VR environment following removal of freezing periods (n = 87 ROIs from 27 sessions in 17 mice). (**c**) Mean normalized $F$ of all LC ROIs (bottom, n = 87) aligned to the switch to the novel environment. To define a baseline and 95% CI (gray shaded region), 1000 shuffles were created from the calcium traces and down-sampled to match the sample size and averaged. This was repeated 1000 times and the mean and 95% CI of this shuffled data were determined for each frame. Red lines indicate periods where two or more consecutive frames passed above the 95% CI of the shuffled baseline. The normalized $F$ of all LC ROIs binned by lap (**d**) or into 50 frame bins (**e**). The baseline and 95% CI (gray shaded region) were defined using the same method as in (**c**). Red lines indicate bins above the baseline 95% CI. Population average position binned (**f**), velocity binned (**g**), and motion onset aligned (**h**) $\Delta F/F \pm$ s.e.m. in LC ROIs (n = 87) in the final laps of the familiar environment (light blue), and the first (orange), third (dark blue), and final laps (green) of the novel environment. Linear regression, $F$ test, position binned (**f**) fam last, p < 1e − 4, nov 1, p < 1e − 68, nov 3, p = 0.48, nov last, p < 0.001; velocity binned (**g**) fam last, p < 1e − 31, nov 1, p < 0.05, nov 3, p < 1e − 5, nov last, $P$ < 1e − 13; motion onset aligned (**h**) fam last, p < 1e − 29, nov 1, p < 0.001, nov 3, p < 1e − 9, nov last, p < 1e − 18. The slope of each lap was compared to the final familiar laps using a one-way ANCOVA with Tukey HSD post hoc test. **p < 0.001, ***$P$ < 1e − 4.

The online version of this article includes the following source data for figure 4:

**Source data 1.** Fluorescence data of locus coeruleus (LC) axons during good behavior in novel virtual reality (VR) environments.

that the relationships between LC activity and behavior in the novel environment quickly return to those in the familiar environment. Interestingly, the slope is significantly increased in the final lap of the novel environment (*Figure 4f*), potentially indicating the development of activity at the novel reward location as has been previously reported (*Kaufman et al., 2020*). Altogether, examining the lap-by-lap dynamics of the position, velocity, and motion onset activity indicates that environmental novelty induces a sharp increase in LC input activity during the first two laps in the novel environment while also inducing a change in behavior that leads to increased LC input activity in subsequent laps.

We also examined the activity of a subset of LC axons (50 ROIs from 11 sessions in 9 mice) during the transition from darkness to the familiar VR environment. These axons showed elevated activity for 20 s following initial exposure to the familiar environment (*Figure 3—figure supplement 2a*). While this activity did not persist as long as the activity following exposure to the novel environment, it

indicates these axons may be generally responsive to abrupt and unpredicted changes to the animal's environment. This fits with the proposed role of the LC in arousal, with novelty driving greater arousal than abrupt exposure to a familiar environment.

## Discussion

During spatial navigation in a familiar environment, activity of VTA DA inputs to dCA1 was strongly modulated by position relative to reward, ramping up as mice approached the end of the track where reward was located. This activity could be related to the animal's distance or time from the rewarded location. Further experiments should be conducted to distinguish between time and distance such as those conducted in VTA DA soma recordings (*Kim et al., 2020*). We have previously shown that this activity is dependent on the history of reward delivery and reflects the animals' reward expectation (*Krishnan et al., 2022*). VTA axons in dCA1 also showed decreasing activity during rest prior to motion onset. However, after removing reward, VTA axons showed no activity prior to motion onset indicating this relationship was driven by reward related activity. This ramping activity cannot be explained by changes in the animals' sensory experience, as the VR environment and waterspout position remained unchanged in the unrewarded condition. Additionally, on the initial laps with no reward, the ramping activity is still present (*Krishnan et al., 2022*) indicating this activity is not directly related to the delivery of water but is instead caused by the animal's internal state of reward expectation.

On average, VTA axons showed activity modulated by velocity in a familiar rewarded environment. This relationship was largely due to the activity of two VTA axons that were strongly modulated by velocity, suggesting that there is heterogeneity in the population of VTA axons in dCA1. A positive relationship between average VTA axon activity and velocity persisted in the unrewarded condition and in the novel environment, indicating velocity encoding in these inputs that is not a result of reward related activity. This heterogeneity in the encoding across individual VTA axons is consistent with studies demonstrating heterogeneous encoding of behavioral variables in VTA DA cell bodies, including activity related to rewards and kinematics (*Engelhard et al., 2019*).

Our findings that VTA DA axons show no novelty-induced activity and instead show reduced activity following exposure to a novel environment, is in contrast with several studies showing novelty-induced activity in VTA DA cell bodies (*Takeuchi et al., 2016*; *Duszkiewicz et al., 2019*; *Lisman and Grace, 2005*), indicating potential heterogeneity in VTA neurons in response to novelty. It is possible that some VTA DA inputs to dCA1 respond to novel environments, and the small number of axons recorded here are not representative of the whole population. Another possibility is that the lack of a novelty response we observe is due to differences in experimental design. Here, mice learned to approach a location for reward which has been shown to lead to ramping activity in dopaminergic VTA neurons (*Howe et al., 2013*; *Krishnan et al., 2022*; *Kim et al., 2020*; *London et al., 2018*; *Jeong et al., 2022*). Following exposure to novelty, the disappearance of reward related activity could obscure any novelty-induced increases in activity. In addition, as noted above, on average we did observe a velocity-associated signal in VTA axons. When mice were exposed to the novel environment their velocity initially decreased. This would be expected to reduce the average signal across the VTA axon population relative to the higher velocity in the familiar environment. It is possible that this decrease could somewhat mask a subtle novelty-induced signal in VTA axons. Therefore, additional experiments should be conducted to investigate the heterogeneity of these axons and their activity under different experimental conditions during tightly controlled behavior.

LC axons showed no position encoding. Instead, they were modulated by velocity and ramped up in activity prior to motion initiation, consistent with recordings of LC axons by others in dCA1 (*Kaufman et al., 2020*) and in the cortex (*Reimer et al., 2016*), respectively. An important question is how is this LC axon activity impacting hippocampal neurons during navigation in a familiar environment? It is possible that LC axons during navigation provide increases in excitability that promotes place cell activity as both dopamine and norepinephrine in the hippocampus can impact cell excitability (*Segal et al., 1991*; *Edelmann and Lessmann, 2011*; *Edelmann and Lessmann, 2018*). Additionally, place cells are flexible during spatial navigation with new place fields forming in familiar environments (*Sheffield et al., 2017*; *Dong et al., 2021*) and shifting position with time/experience (*Dong et al., 2021*). Place fields can also shift to follow changing reward (*Gauthier and Tank, 2018*) and object locations (*Bourboulou et al., 2019*). Dopamine and norepinephrine have also been shown to impact hippocampal synaptic plasticity (*Zhang et al., 2009*; *Edelmann and Lessmann, 2011*; *Hagena and*

*Manahan-Vaughan, 2012*; *Goh and Manahan-Vaughan, 2013*). Therefore, LC inputs may promote the plasticity necessary for place cells to flexibly adapt to changes in a familiar environment (*Kaufman et al., 2020*; *Redondo and Morris, 2011*). In other words, LC inputs could allow the hippocampus to be flexible during navigation through their impacts on synaptic plasticity (*Takeuchi et al., 2016*; *Yamasaki and Takeuchi, 2017*; *Duszkiewicz et al., 2019*).

Exposure to environmental novelty leads to an increase in dopamine in the dorsal hippocampus (*Ihalainen et al., 1999*) and promotes synaptic plasticity (*Li et al., 2003*; *Hagena and Manahan-Vaughan, 2012*), hippocampal replay (*McNamara et al., 2014*; *Dupret et al., 2010*), and memory persistence (*Li et al., 2003*; *Cohen et al., 2017*). In our experiment, exposure to a novel environment caused an increase in LC axon activity but not in VTA DA axon activity, supporting findings that novel experiences induce activity of LC neurons (*Takeuchi et al., 2016*). As discussed above, the slowing down of animal behavior in the novel environment could have decreased LC axon activity and reduced the magnitude of the novelty signal we detected during running. The novelty signal we report here may therefore be an under estimate of its magnitude under matched behavioral settings. The increased activity of LC neurons in the novel environment could increase hippocampal neuron activity (*Wagatsuma et al., 2018*), increase efficacy of Schaffer Colateral synapses (*Takeuchi et al., 2016*), stabilize place cells across days (*Wagatsuma et al., 2018*), and enhance memory persistence (*Wagatsuma et al., 2018*; *Takeuchi et al., 2016*; *Chowdhury et al., 2022*) through dopamine receptor-dependent mechanisms. Importantly, while LC inputs to CA1 have been shown to cause an increase in activity (*Wagatsuma et al., 2018*; *Chowdhury et al., 2022*), shape over-representation of novel reward locations (*Kaufman et al., 2020*), and modulate memory linking (*Chowdhury et al., 2022*), they have not been shown to play a role in the formation of contextual memories (*Chowdhury et al., 2022*) or stabilization of place cell maps across days (*Wagatsuma et al., 2018*) in dCA1. However, it is possible this novelty-induced activity in LC inputs to dCA1 impacts the formation of instant place fields observed in novel environments (*Sheffield et al., 2017*; *Dong et al., 2021*). Instant place fields form on the first lap of a novel environment, right when the LC novelty signal is highest in dCA1, suggesting LC inputs may play a role in their formation or stabilization. Therefore, further experiments should investigate the role of LC axons on dCA1 place fields on a trial-by-trial basis.

While LC neurons have been shown to impact novelty encoding through dopaminergic mechanisms (*Wagatsuma et al., 2018*; *Takeuchi et al., 2016*; *Chowdhury et al., 2022*), this does not exclude the possibility that they also release norepinephrine during exposure to novelty and exploration of a familiar environment. Indeed, hippocampal levels of norepinephrine also increase during exposure to environmental novelty (*Lima et al., 2019*; *Moreno-Castilla et al., 2017*), but how this norepinephrine release effects hippocampal function is not well understood. Additionally, it is not known whether norepinephrine and dopamine are released from the same LC inputs or from distinct sets of LC inputs. Dopamine is in the synthesis pathway of norepinephrine and in LC neurons it is loaded into vesicles where it is then converted to norepinephrine by dopamine $\beta$-hydroxylase (*Cimarusti et al., 1979*). It is possible that high levels of activity of LC inputs, like those occurring during exposure to novelty, lead to release of vesicles before dopamine can be converted to norepinephrine thus leading to the release of dopamine under these conditions. However, low levels of LC activation, like those observed during familiar environment exploration, may provide time for dopamine to be converted to norepinephrine and thus lead to the release of norepinephrine from LC terminals under these conditions. Further experiments investigating the dynamics of dopamine conversion and release from LC terminals in the hippocampus should be conducted to test this hypothesis.

Here, we show that LC input activity is modulated by velocity, time to motion onset, abrupt exposure to a familiar environment and abrupt exposure to novel environments. Each of these conditions is associated with an increase in arousal, and LC activity has been strongly linked to arousal levels (*Aston-Jones and Bloom, 1981*; *Berridge and Waterhouse, 2003*; *Carter et al., 2010*). Therefore, rather than encoding each of these variables independently, LC inputs are likely encoding the animals arousal level during spatial navigation. It has been shown that attention and arousal levels impact tuning properties in many cortical areas and this is thought to be mediated through LC activity (*Shulman et al., 1979*; *Bouret and Sara, 2002*; *Martins and Froemke, 2015*; *Waterhouse and Navarra, 2019*). Similarly, changes in the animals' brain state, including changes in attention (*Kentros et al., 2004*) and engagement (*Pettit et al., 2022*), alter the tuning properties of place cells. This

indicates arousal could impact the function of hippocampal neurons through these LC inputs, either directly or through astrocytes (*Rupprecht et al., 2023*).

The distinct activity dynamics exhibited by LC and VTA DA axons during spatial navigation of familiar and novel environments underscore their distinct contributions to hippocampal-dependent learning and memory processes. Notably, these findings reinforce the notion that VTA DA inputs play a pivotal role in the ongoing maintenance and updating of associations between expected rewards and the locations that lead to them, while LC axons appear to be integral to the process of encoding memories of entirely new environments and stimuli.

## Methods
### Subjects
All experimental and surgical procedures were in accordance with the University of Chicago Animal Care and Use Committee guidelines. For this study, we used 10- to 20-week-old male *Slc6a3* Cre+/− (DAT-Cre+/−) mice obtained from JAX labs and *Slc6a2* Cre+/− (NET-Cre+/−) (23–33 g) mice obtained from the Tonegawa lab (*Wagatsuma et al., 2018*). Male mice were used over female mice due to the size and weight of the headplates (9.1 mm × 31.7 mm, 2 g) which were difficult to firmly attach on smaller female skulls. Mice were individually housed in a reverse 12 hr light/dark cycle at 72°F and 47% humidity, and behavioral experiments were conducted during the animal's dark cycle.

### Mouse surgery and viral injections
Mice were anesthetized (1–2% isoflurane) and injected with 0.5 ml of saline (intraperitoneal injection) and 0.05 ml of Meloxicam (1–2 mg/kg, subcutaneous injection) before being weighed and mounted onto a stereotaxic surgical station (David Kopf Instruments). A small craniotomy (1–1.5 mm diameter) was made over the VTA (±0.5 mm lateral, 3.1 mm caudal of Bregma) of DAT-Cre+/− mice or over the locus coeruleus (LC) (±0.875 mm lateral, −5.45 mm caudal of Bregma). The genetically encoded calcium indicator, pAAV-hsyn-Flex-Axon-GCaMP6s (pAAV-hSynapsin1-FLEx-axon-GCaMP6s was a gift from Lin Tian (Addgene viral prep #112010-AAV5; http://n2t.net/addgene:112010; RRID:Addgene_112010)) was injected into the VTA of DAT-Cre+/− mice (200 nl at a depth of 4.4 mm below the surface of the dura) or the LC of NET-Cre+/− mice (200 nl at a depth of 3.65 mm below Bregma). For a subset (4/8) of VTA recordings, a different GCaMP variant, pAAV-Ef1A-Flex-Axon-GCaMP7b, was injected due to the difficulty finding and recording VTA axons in dCA1 (pAAV-Ef1a-Flex-Axon-GCaMP7b) (pAAV-Ef1a-Flex-Axon-GCaMP7b was a gift from Rylan Larsen – Addgene plasmid #135419; http://n2t.net/addgene:135419; RRID: Addgene 135419). Following injections, the site was covered up using dental cement (Metabond, Parkell Corporation) and a metal headplate (9.1 mm × 31.7 mm, Atlas Tool and Die Works) was also attached to the skull with the cement. To reduce bleeding and swelling during the hippocampal window implantation, mice were separated into individual cages and water restricted for 3 weeks (0.8–1.0 ml/day). Mice then underwent surgery to implant a hippocampal window as previously described (*Dombeck et al., 2010*). Following implantation, the headplate was attached with the addition of a head-ring cemented on top of the headplate which was used to house the microscope objective and block out ambient light. Post-surgery mice were given 2–3 ml of water/day for 3 days to enhance recovery before returning to the reduced water schedule (0.8–1.0 ml/day).

### Behavior and VR
Our VR and treadmill setup were designed similar to previously described setups (*Sheffield et al., 2017*; *Heys et al., 2014*). The virtual environments that the mice navigated through were created using VIRMEn (*Aronov and Tank, 2014*). 2 m (DAT-Cre mice) or 3 m (NET-Cre mice) linear tracks rich in visual cues were created that evoked numerous place fields in mice as they moved along the track at all locations (*Figure 1*; *Bourboulou et al., 2019*). Mice were head restrained with their limbs comfortably resting on a freely rotating styrofoam wheel (treadmill). Movement of the wheel caused movement in VR by using a rotary encoder to detect treadmill rotations and feed this information into our VR computer, as in *Sheffield et al., 2017*; *Heys et al., 2014*. Mice received a water reward (4 μl) through a waterspout positioned directly in front of the animal's mouth upon completing each traversal of the track (a lap), which was associated with a clicking sound from the solenoid. Licking was monitored by a capacitive sensor attached to the waterspout. Upon receiving the water reward,

a short VR pause of 1.5 s was implemented to allow for water consumption and to help distinguish laps from one another rather than them being continuous. Mice were then virtually teleported back to the beginning of the track and could begin a new traversal. Mouse behavior (running velocity, track position, reward delivery, and licking) was collected using a PicoScope Oscilloscope (PICO4824, Pico Technology, v6.13.2). Behavioral training to navigate the virtual environment began 4–7 days after window implantation (30 min/day) and continued until mice reached >2 laps/min, which took 10–14 days (although some mice never reached this threshold). Mice that reached this behavioral threshold and had adequate GCaMP expression were imaged the following day. Out of the 36 NET-Cre mice injected, 15 were never recorded for either failing to reach behavioral criteria, or a lack of visible expression in axons. Out of the 54 DAT-Cre mice injected, images were never conducted in 36 for lack of expression or failing to reach behavioral criteria.

## Two-photon imaging

Imaging was done using a laser scanning two-photon microscope (Neurolabware). Using a 8-kHz resonant scanner, images were collected at a frame rate of 30 Hz with bidirectional scanning through a 16×/0.8 NA/3 mm WD water immersion objective (MRP07220, Nikon). GCaMP6s and GCaMP7b were excited at 920 nm with a femtosecond pulsed two photon laser (Insight DS + Dual, Spectra-Physics) and emitted fluorescence was collected using a GaAsP PMT (H11706, Hamamatsu). The average power of the laser measured at the objective ranged between 50–80 mW. A single imaging field of view between 400 and 700 μm equally in the *x/y* direction was positioned to collect data from as many VTA or LC axons as possible. Time-series images were collected from 3 to 5 planes spaced 2 μm apart using an electric lens to ensure axons remained in a field of view and reduce power going to an individual plane. Images were collected using Scanbox (v4.1, Neurolabware) and the PicoScope Oscilloscope (PICO4824, Pico Technology, v6.13.2) was used to synchronize frame acquisition timing with behavior.

## Imaging sessions

The familiar environment was the same environment that the animals trained in. The experiment protocol for single day imaging sessions is shown in *Figure 1*. Each trial lasted 8–12 min and was always presented in the same order. Mice were exposed to the familiar rewarded environment for 10 min, then were immediately teleported to the start of a novel-rewarded VR environment and allowed to navigate for 10 min. Mice on average ran 19 ± 3.8 (mean ± 95% CI) laps in the familiar environment, at which point they were teleported to the novel environment and imaging continued for 30 ± 5.4 laps. The Novel-rewarded environment (N) had distinct visual cues, colors, and visual textures, but the same dimensions (2 or 3 m linear track) and reward location (end of the track) as the familiar environment. Imaging sessions with large amounts of drift or bleaching were excluded from analysis (eight sessions for NET mice, six sessions for LC mice). Sample size for NET-Cre mice was based off previous experiments examining calcium activity of individual somas and axons (*Reimer et al., 2016*; *Engelhard et al., 2019*). For DAT-Cre mice sample size was largely limited by the difficulty of recordings and fields of view only containing one axon.

## Histology and brain slices imaging

We checked the expression axon-GCaMP to confirm expression was restricted to the VTA of DAT-Cre mice and the LC of NET-Cre. Mice were deeply anesthetized with isoflurane and perfused with 10 ml phosphate-buffered saline (PBS) followed by 20 ml 4%paraformaldehyde in PBS. The brains were removed and immersed in 30% sucrose solution overnight before being sectioned at 30-μm thickness on a cryostat. Brain slices were collected into well plates containing PBS. Slices were washed five times with PBS for 5 min then were blocked in 1% bovine serum albumin, 10% Normal goat serum, and 0.1% Triton X-100 for 2 hr. Brain slices were then incubated with 1:500 rabbit-$\alpha$-TH (MAB318, Sigma-Aldrich) and 1:500 mouse-$\alpha$-GFP (SAB2702197, Sigma-Aldrich)in blocking solution at 4°C. After 48 hr, the slices were incubated with 1:1000 goat-$\alpha$-rabbit Alexa Fluor 647 nm secondary antibody (A32731, Thermo Fisher) and 1:1000 goat-$\alpha$-mouse Alexa Fluor 488 nm (A32723, Thermo Fisher) for 2 hr. Brain slices were then collected on glass slides and mounted with a mounting media with 4'6-diamidino-2-phenylindole (DAPI; SouthernBiotech DAPI-Fluoromount-G Clear Mounting Media, 010020). The whole-brain slices were imaged under ×10 and ×40 with a Caliber I.D. RS-G4 Large

Format Laser Scanning Confocal microscope from the Integrated Light Microscopy Core at the University of Chicago.

## Image processing and ROI selection

Time-series images were preprocessed using Suite2p (v0.10.1)[79]. Movement artifacts were removed using rigid and non-rigid transformations and assessed to ensure absence of drifts in the *z*-direction. Datasets with *z*-drifts were discarded, as determined by visually assessing imaging sessions, followed by using the registration metrics output by suite2p (eight sessions). For axon imaging, ROIs were first defined using Suite2p and manually inspected for accuracy. ROIs were then hand drawn over all segments of Suite2p defined active axons using ImageJ to ensure all axon segments were included for analysis. Fluorescent activity for each ROI was extracted and highly correlated ROIs (Pearson correlation coefficient ≥0.7) were combined and their fluorescent activity was extracted. To be included Baseline corrected $\Delta F/F$ traces across time were then generated for each ROI using both a small window of 300 frames for lap-by-lap analysis, and a larger sliding window of 2000 frames to avoid flattening slow signals for novelty response analysis. Additional ROIs were drawn over autofluorescent structures that were not identified by suite2p. These 'blebs' were processed in the same way as axon ROIs and used as controls to check for imaging and motion artifacts.

To remove low signal-to-noise axons, we defined the SNR of each ROI using the power spectrum of their fluorescent activity similar to *Reimer et al., 2016*. For frequencies above 1 Hz, the power was defined as noise because this sits outside of the range of frequencies possible for GCaMP6s fluorescence. The SNR ratio was then defined as the ratio of the peak power between 0.5 and 1 Hz over the average power between 1 and 3 Hz. The SNR of 'blebs' was also determined and any axon with an SNR greater than 1.5 std from the mean of the 'blebs' SNR was used for analysis (110/231 LC ROIs, 9/9 VTA ROIs).

Additionally, it was observed that a subset of axon ROIs would greatly increase fluorescence at seemingly random time points and remain elevated for the rest of the trial. This activity could be due to the axons being unhealthy and filling with calcium. Therefore, we identified these axons using the *cusum* function in matlab to detect changes in mean activity that remained elevated for at least 2000 frames or at least 500 frames if they were still elevated at the end of the recording session and removed them from analysis (23/110 LC ROIs, 0/7 VTA ROIs).

## Behavioral analysis

Mouse velocity was calculated as the change in VR position divided by the sampling rate and smoothed using a Savitzky-Golay filter with a 7 frame window and 5 degree polynomial. To find the lap mean velocity, periods where the mice were immobile (velocity <5 cm/s) were removed and the average velocity during the remaining frames was calculated. The lap mean freezing ratio was calculated as the number of frames spent immobile (velocity <5 cm/s) divided by the total number of frames for each lap.

## Axon imaging analysis

For the three measures below, to avoid weighting axons with a high SNR more than others each ROI was normalized by $(\Delta F/F - \Delta F/Fmin)/(\Delta F/Fmax - \Delta F/Fmin)$ where $\Delta F/Fmin$ is the 1st quantile and $\Delta F/Fmax$ was the 99 quantile for each ROI. The 1st and 99th quantiles were used in order to avoid normalize to noisy outlier data points.

## Position binned fluorescence

To find the position binned fluorescent activity of each ROI, the track was divided into 5 cm bins. For each lap, the average fluorescence in each bin was calculated for each ROI. The position binned fluorescence was then averaged across all laps in each environment to find the mean position binned activity in the familiar and novel environments.

## Velocity binned fluorescent activity

To find the velocity binned fluorescent activity for each ROI, the velocity was divided into 1 cm/s bins from 1 to 30 cm/s. Velocities above this 14 cm/s were excluded from figures because not all mice ran faster than 14 cm/s. For each lap, the ROIs average fluorescence in each velocity bin was calculated

and then averaged across all laps in each environment to find the velocity binned activity in the familiar and novel environments.

### Motion initiation aligned fluorescence

Periods where mice were immobile (velocity <5 cm/s) for at least 1.5 s then proceed to run (velocity ≥5 cm/s) for at least 3 s were identified. The fluorescent activity for ROIs for these periods was aligned to the frame mice began running (velocity crossed above 5 cm/s). The average aligned fluorescent activity of each ROI was then determined for each environment.

### Linear regression analysis

To assess dynamics between each of the above measures and calcium activity of LC and VTA axons, we performed linear regression on the population's familiar environment data and significance was assessed with an F test. To compare the dynamics between LC and VTA axons, we performed exact testing based on Monte-Carlo resampling (1000 resamples with sample size matching the lower sample size condition) as detailed in legends (*Figure 2e*).

To assess the changing position and velocity encoding of LC axons following exposure to a novel environment, we performed linear regression on the population fluorescence data of the average of the last four laps in the familiar environment, and each of the first three laps in the novel environment for each measure. The significance for the fit of each line was assessed with an *F* test, and an ANCOVA was conducted to test for differences in slope between the four laps. The same process was conducted for the motion initiation dynamics, but only using ROIs in mice who paused within the first two laps and 30 s following exposure to the novel environment.

### Novel response analysis

To examine the response of LC and VTA axons to the novel VR environment, the fluorescence data were normalized by the mean for each ROI and aligned to the frame where the mice were switched to the novel environment and the mean normalized *F* for LC and VTA ROIs at each time point was calculated. Baseline fluorescent activity was then calculated for LC and VTA ROIs separately by generating 1000 shuffled traces of the ROIs calcium activity and subsampling down to the sample size (90 for LC; 7 for VTA) 1000 times and finding the mean of the subsampled shuffles. The mean and 95% CI of all 1000 subsamples were found and the mean activity of LC and VTA ROIs was considered significantly elevated when it passed above the 95% CI of the shuffled data. The same process was repeated to define a baseline for the time binned data (fluorescent activity divided into 50 frame bins) and the lap binned data (mean activity for each lap).

Additionally, to account for changes in behavior between the familiar and novel environments, periods where the animals were immobile (velocity ≤0.2 cm/s) were removed and running periods were concatenated together and aligned to the switch to the novel environment. Here, we again defined a baseline for the time mean traces, time binned activity, and the lap binned activity using the above bootstrapping approach.

## Acknowledgements

We thank Dr. Seetha Krishnan and Dr. Douglas Goodsmith for feedback on the manuscript. We also thank Dr. Antoine Madar for extensive and helpful discussions on using appropriate statistical analysis used in this paper. This work was supported by The Whitehall Foundation, The Searle Scholars Program, The Sloan Foundation, The University of Chicago Institute for Neuroscience start-up funds, the NIH 1DP2NS111657-01, and the NIH BRAIN Initiative RF1NS127123 awarded to MS and a T32 training grant T32DA043469 from National Institute on Drug Abuse awarded to CH.

## Additional information

### Funding

| Funder | Grant reference number | Author |
| --- | --- | --- |
| BRAIN Initiative | 1RF1NS127123-01 | Mark Sheffield |
| National Institute of Neurological Disorders and Stroke | 1DP2NS111657-01 | Mark Sheffield |
| Whitehall Foundation | | Mark Sheffield |
| Searle Scholars Program | | Mark Sheffield |
| Alfred P. Sloan Foundation | | Mark Sheffield |
| The University of Chicago Institute for Neuroscience startup funds | | Mark Sheffield |
| National Institute on Drug Abuse | T32DA043469 | Chad Heer |

The funders had no role in study design, data collection, and interpretation, or the decision to submit the work for publication.

### Author contributions

Chad Heer, Data curation, Formal analysis, Methodology, Writing - original draft, Writing - review and editing; Mark Sheffield, Conceptualization, Resources, Supervision, Funding acquisition, Validation, Methodology, Writing - original draft, Writing - review and editing

### Author ORCIDs

Chad Heer ⓘ https://orcid.org/0000-0002-4527-0405
Mark Sheffield ⓘ https://orcid.org/0000-0003-0969-7820

### Ethics

All experimental and surgical procedures were approved by the University of Chicago Animal Care and Use Committee under protocol number: 72508 and carried out in strict accordance with their guidelines. All surgery was performed under isoflurane anesthesia, and every effort was made to minimize suffering.

Reviewer #1 (Public review): https://doi.org/10.7554/eLife.95213.4.sa1
Reviewer #2 (Public review): https://doi.org/10.7554/eLife.95213.4.sa2
Reviewer #3 (Public review): https://doi.org/10.7554/eLife.95213.4.sa3
Author response https://doi.org/10.7554/eLife.95213.4.sa4

---

## Additional files

### Supplementary files
• MDAR checklist

### Data availability

Each figure in the manuscript has an associated source data file that contains the numerical data used to generate each figure panel. Time-series behavioral data and imaging data shown in figures can be found here: https://doi.org/10.5061/dryad.ffbg79d4h. Scripts used for data analysis are available on Github (https://github.com/chadheer/LC_VTA_paper; copy archived at *Heer, 2024*).

The following dataset was generated:

| Author(s) | Year | Dataset title | Dataset URL | Database and Identifier |
|-----------|------|---------------|-------------|------------------------|
| Heer CM, Sheffield MEJ | 2024 | Distinct catecholaminergic pathways projecting to hippocampal CA1 transmit contrasting signals during navigation in familiar and novel environments | https://doi.org/10.5061/dryad.ffbg79d4h | Dryad Digital Repository, 10.5061/dryad.ffbg79d4h |

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
