## [Editor Report · eLife assessment]

This study presents **important** findings on the differential activity of noradrenergic and dopaminergic input to dorsal hippocampus CA1 in head-fixed mice traversing a runway in a virtual environment that is familiar or novel. The data are rigorously analysed, and the observed divergence in the dynamics of activity in the dopaminergic and noradrenergic axons is **solid**. Future studies, using specific manipulations of the two distinct midbrain inputs combined with behavioral testing, are required to strengthen the claim that distinct signals to the hippocampus cause distinct behavioral effects.

---

## [Referee Report · Reviewer #1 (Public review)]

Summary:

Heer and Sheffield used 2 photon imaging to dissect the functional contributions of convergent dopamine and noradrenaline inputs to the dorsal hippocampus CA1 in head restrained mice running down a virtual linear path. Mice were trained to collect water reward at the end of the track and on test days, calcium activity was recorded from dopamine (DA) axons originating in ventral tegmental area (VTA, n=7) and noradrenaline axons from the locus coeruleus (LC, n=87) under several conditions. When mice ran laps in a familiar environment, VTA DA axons exhibited ramping activity along the track that correlated with distance to reward and velocity to some extent, while LC input activity remained constant across the track, but correlated invariantly with velocity and time to motion onset. A subset of recordings taken when the reward was removed showed diminished ramping activity in VTA DA axons, but no changes in the LC axons, confirming that DA axon activity is locked to reward availability. When mice were subsequently introduced to a new environment, the ramping to reward activity in the DA axons disappeared, while LC axons showed a dramatic increase in activity lasting 90s (6 laps) following the environment switch. In the final analysis, the authors sought to disentangle LC axon activity induced by novelty vs. behavioral changes induced by novelty by removing periods in which animals were immobile, and established that the activity observed in the first 2 laps reflected novelty-induced signal in LC axons.

The revised manuscript included additional evidence of increased (but transient) signal in LC axons after a transition to a novel environment during periods of immobility, and also that a change from dark to familiar environment induces a peak in LC axon activity, showing that LC input to dCA1 may not solely signal novelty.

Strengths:

The results presented in this manuscript provide insights into the specific contributions of catecholaminergic input to the dorsal hippocampus CA1 during spatial navigation in a rewarded virtual environment, offering a detailed analysis at the resolution of single axons. The data analysis is thorough and possible confounding variables and data interpretation are carefully considered.

Weaknesses:

Aspects of the methodology, data analysis, and interpretation diminish the overall significance of the findings, as detailed below.

The LC axonal recordings are well powered, but the DA axonal recordings are severely underpowered, with recordings taken from a mere 7 axons (compare to 87 LC axons). Additionally, 2 different calcium indicators with differential kinetics and sensitivity to calcium changes (GCaMP6S and GCaMP7b) were used (n=3, n=4 respectively) and the data pooled. This makes it very challenging to draw any valid conclusions from the data, particularly in the novelty experiment. The surprising lack of novelty-induced DA axon activity may be a false negative. Indeed, at least 1 axon (axon 2) appears to be showing novelty-induced rise in activity in Figure 3C. Changes in activity in 4/7 axons are also referred to as a 'majority' occurrence in the manuscript, which again is not an accurate representation of the observed data

The authors conducted analysis on recording data exclusively from periods of running in the novelty experiment to isolate the effects of novelty from novelty-induced changes in behavior. However, if the goal is to distinguish between changes in locus coeruleus (LC) axon activity induced by novelty and those induced by motion, analyzing LC axon activity during periods of immobility would enhance the robustness of the results.

The authors attribute the ramping activity of the DA axons to the encoding of the animals' position relative to reward. However, given the extensive data implicating the dorsal CA1 in timing, and the remarkable periodicity of the behavior, the fact that DA axons could be signalling temporal information should be considered.

The authors should explain and justify the use of a longer linear track (3m, as opposed to 2m in the DAT-cre mice) in the LC axon recording experiments.

AFTER REVISIONS:

The authors have addressed my concerns in a thorough manner. The reviewer also appreciates the increased transparency of reporting in the revised manuscript.

Listed below are some remaining comments.

The increase in LC activity with any change in environment (from familiar to novel or from dark to familiar) suggests that LC input acts not solely as a novelty signal, but as a general arousal or salience signal in response to environmental changes. Based on this, I have a couple of questions:

• Is the overall claim that LC input to the dHC signals novelty still valid based on observed findings - as claimed throughout the manuscript?

• Would the omission of a reward be considered a salient change in the environment that activates LC signals, or is the LC not involved with processing reward-related information? Has the activity of LC and VTA axons been analysed in the seconds following reward presentation and/or omission?

---

## [Referee Report · Reviewer #2 (Public review)]

Summary:

The authors used 2-photon Ca2+-imaging to study the activity of ventral tegmental area (VTA) and locus coeruleus (LC) axons in the CA1 region of the dorsal hippocampus in head-fixed male mice moving on linear paths in virtual reality (VR) environments.

The main findings were as follows:

- In a familiar environment, activity of both VTA axons and LC axons increased with the mice's running speed on the Styrofoam wheel, with which they could move along a linear track through a VR environment.

- VTA, but not LC, axons showed marked reward position-related activity, showing a ramping-up of activity when mice approached a learned reward position.

- In contrast, activity of LC axons ramped up before initiation of movement on the Styrofoam wheel.

- In addition, exposure to a novel VR environment increased LC axon activity, but not VTA axon activity.

Overall, the study shows that the activity of catecholaminergic axons from VTA and LC to dorsal hippocampal CA1 can partly reflect distinct environmental, behavioral and cognitive factors. Whereas both VTA and LC activity reflected running speed, VTA, but not LC axon activity reflected approach of a learned reward and LC, but not VTA, axon activity reflected initiation of running and novelty of the VR environment.

I have no specific expertise with respect to 2-photon imaging, so cannot evaluate the validity of the specific methods used to collect and analyse 2-photon calcium imaging data of axonal activity.

Strengths:

(1) Using a state-of-the-art approach to record separately the activity of VTA and LC axons with high temporal resolution in awake mice moving through virtual environments, the authors provide convincing evidence that activity of VTA and LC axons projecting to dorsal CA1 reflect partly distinct environmental, behavioral and cognitive factors.

(2) The study will help (a) to interpret previous findings on how hippocampal dopamine and norepinephrine or selective manipulations of hippocampal LC or VTA inputs modulate behavior and (b) to generate specific hypotheses on the impact of selective manipulations of hippocampal LC or VTA inputs on behavior.

Comments on revised version:

I thank the authors for including a sample size justification.

The justification is based on previous studies using similar sample sizes to characterize behavioral correlates of LC and VTA activity and on practical reasons. I note that to improve reproducibility, it would be preferable to have predefined target sample sizes based on predefined plans for statistical analysis.

---

## [Referee Report · Reviewer #3 (Public review)]

Summary:

Heer and Sheffield provide a well-written manuscript that clearly articulates the theoretical motivation to investigate specific catecholaminergic projections to dorsal CA1 of the hippocampus during a reward-based behavior. Using 2-photon calcium imaging in two groups of cre transgenic mice, the authors examine activity of VTA-CA1 dopamine and LC-CA1 noradrenergic axons during reward seeking in a linear track virtual reality (VR) task. The authors provide a descriptive account of VTA and LC activities during walking, approach to reward, and environment change. Their results demonstrate LC-CA1 axons are activated by walking onset, modulated by walking velocity, and heighten their activity during environment change. In contrast, VTA-CA1 axons were most activated during approach to reward locations. Together the authors provide a functional dissociation between these catecholamine projections to CA1. A major strength to their approach is the methodological rigor of 2-photon recording, data processing, and analysis approaches to accommodate their unequal LC-CA1 and VTA-CA1 sample sizes. These important systems neuroscience studies provide solid evidence that will contribute to the broader field of navigation and memory.

Weaknesses:

The conclusions of this manuscript are mostly well supported by the data. However, increasing the sample size of the VTA-CA1 group and using experimental methods that are identical among LC-CA1 and VTA-CA1 groups would help to fully support the author's conclusions.

---

## [Author Response]

The following is the authors’ response to the previous reviews.

**Recommendations for the authors:**

**Reviewer #1 (Recommendations For The Authors):**
Please reorder the supplementary figures in the order they are referred to in the Results section for ease of reading. Supp Fig 5 b - should read 'Mean normalized fluorescence of LC ROIs (n = 87) during immobile periods aligned to the switch from familiar to novel environment.’

We thank the reviewer for highlighting these issues and have reordered the supplementary figures and edited the figure legends appropriately.

**Reviewer #2 (Recommendations For The Authors):**
The authors should include sample size justifications (e.g. based on previous studies, considerations of statistical power, practical considerations, or a combination of these factors).

In response to this concern, we have added a statement to the “Imaging Sessions” section of the methods. Here we highlight sample sizes were largely based on previous studies and/or limited by the difficulty of recordings and the limited number of visible axons per imaging session.

**Reviewer #3 (Recommendations For The Authors):**
The addition of Supp. Fig 5 partially addresses my previous point 3. However, the claim of dissociation between VTA-CA1 and LC-CA1 would be strengthened by showing that VTA-CA1 axons do not respond to the darkness -> familiar environment in Supp Fig 5. This is particularly important given that (1) the additional 2 VTA-CA1 axons in the revision were not recorded during transitions to novel environments and (2) the overall concern of the reviewers that the low n and heterogeneity of the VTA-CA1 dataset may lead to a false negative. Providing VTA-CA1 data for the darkness -> familiar environment would provide a within-manuscript replication that these axons are not responding to environment changes; a major claim of this manuscript.

While we agree that data of VTA-CA1 axons during the switch from darkness to the familiar environment would provide additional evidence that these axons are not responding to environment changes, unfortunately, VTA axons were not recorded during the switch from familiar to novel.